# Improving Student Learning of Energy Systems through Computational Tool Development Process in Engineering Courses

**Jian Zhang [1], Heejin Cho [2],\* and Pedro J. Mago [3]**

1  Mechanical Engineering, Richard J. Resch School of Engineering, University of Wisconsin-Green Bay, Green Bay, WI 54311, USA; zhangj@uwgb.edu
2  Department of Mechanical Engineering, Mississippi State University, Mississippi State, MS 39762, USA
3  Department of Mechanical and Aerospace Engineering, West Virginia University, Morgantown, WV 26506, USA; pedro.mago@mail.wvu.edu
\*  Correspondence: cho@me.msstate.edu

**Abstract:** Advancements in computer and mobile technologies have driven transformations of classroom activities in engineering education. This evolution provides instructors more opportunities to introduce computational tools that can be effectively used and promoted in engineering education to advance students' learning process when the tools are appropriately utilized in the classroom activities. This paper presents a methodology to improve student learning of energy systems through a class assignment implementing a self-developed computational tool using Microsoft Excel and utilizing the tool to enhance their learning experience. The proposed method, a student-centered learning approach, was applied in a technical elective course called "Power Generation Systems" within a mechanical engineering curriculum. In the course, students were guided to develop a computational tool by themselves based on their learning of the fundamental principles and governing equations of a thermodynamics cycle. The self-developed computational tool allows the students to focus on more design-oriented problems, instead of the calculation process. Using the self-developed tool, students can have an enhanced understanding of the energy system performance in varying design and operational conditions and can perform the parametric analysis and visualization of essential parameters. Feedback from the students and class instructors proves that the self-development and use of the tool can significantly improve the students' learning experience in the implemented course, make the course more dynamic, and motivate the students to learn the material more iteratively. In addition, students feel confident using computational tools to perform analysis, and are willing to develop more tools for other energy-related engineering applications.

**Keywords:** higher education; computer-based instruction; power generation systems; computational tool; exergy analysis

## 1. Introduction

Advancements in computer and mobile technologies have driven transformations of classroom activities in energy-related engineering education. More frequently, engineering students are required to use their computers and mobile to perform daily class exercises and homework assignments. This evolution also provides instructors more opportunities to introduce computational tools that can be effectively used and promoted in energy-related engineering education to advance student's learning process when the tools are appropriately utilized in classroom activities. However, educators also face challenges to effectively adopt and adapt computer technologies into classrooms, to promote fruitful and meaningful learning [1]. Many researchers and instructors have investigated the effectiveness and advantages of integrating computational tools into engineering courses [1–4]. Effective integration and utilization of computational tools in engineering courses can advance

students' learning in various ways, i.e., (a) improve their understanding of background materials through effective parametric studies [5], (b) obtain problem-solving skills for complex engineering problems [2], (c) eliminate time-intensive and repetitive manual calculations [6], (d) promote an interactive learning environment using computation tools [7], and (e) gain the ability to interpret and verify results using computational tools [8].

Various computational tools have been introduced to demonstrate their effective integration and utilization in engineering education, e.g., Phyton [3,9] (A software program https://www.python.org/), MATLAB [4,9] (A software program https://www.mathworks.com/products/matlab.html), MathCAD [8,10] (A software program https://www.mathworks.com/products/matlab.html), Microsoft Excel [5–7,11,12]. Although advancements in engineering calculation tools have been rapidly evolving, still, spreadsheet-based computational tools are dominantly used in engineering education because of their heavy use in industry [13], the wide accessibility and availability of programs [14], etc. Various advantages of using spreadsheet-based computational tools have been discussed and introduced in different engineering education disciplines, e.g., [11,14,15]. This paper illustrates how an Excel-based spreadsheet tool can be developed by students and integrated into a mechanical engineering curriculum to enhance students' understanding and learning of course materials.

Computational tools have been introduced in energy-related and thermodynamic-based engineering courses and demonstrated that they could have positive impacts on student's learning. Vieira et al. [3] introduced a Python-based software tool to a thermodynamics course and evaluated its effects on the students' learning, with the observation of a positive impact. F. Cruz-Peragon et al. [16] integrated a spreadsheet-based reciprocating engine model to enhance the students' understanding of thermodynamic cycles for different engines. Caretto et al. [17] presented the development of an Excel spreadsheet-based thermodynamic property calculation tool and discussed how their tool was integrated into fundamental thermodynamics courses and design projects in their mechanical engineering curriculum. Mago and Luck [5] demonstrated that MathCAD and Excel spreadsheet-based tools for psychrometric processes in buildings could be effectively integrated into an air conditioning technical elective course to enhance students' learning experience. Knizley and Mago [18,19] demonstrated that utilizing Excel spreadsheet-based tools in the classroom can promote a student-centered learning environment and improve the student understanding of class materials and teaching efficiency in engineer courses.

This paper aims to present a pedagogical method, using a student-centered learning approach, to improve the student learning of energy systems through a class assignment implementing a self-developed computational tool using Microsoft Excel in an energy-related engineering course, called Power Generation Systems. There are three phases to implementing and evaluating the proposed method in this study. In Phase 1, the instructor provides the background information and fundamental knowledge needed to understand the combined cycle systems. In Phase 2, the students develop and implement the computational tool based on their learning from lectures and a prior class assignment, to perform a manual calculation of combined power cycle design and performance at rated conditions. In this phase, the instructor facilitates the student learning and guides the students during the tool development. Finally, in Phase 3, the students take the survey and give the instructor feedback on the effect and benefit of their computational tool development task. The result of this study shows how the proposed method can effectively promote student learning experience and motivation.

## 2. Fundamentals of Combined Power Cycle

This section demonstrates the thermodynamic model and process disseminated to the students to develop their own computation tool of the combined power cycles in a senior-level mechanical engineering course, Power Generation Systems. A schematic of the combined power cycle is illustrated in Figure 1. The combined cycle involves a gas turbine cycle, which uses air as the working fluid, and a Rankine cycle, which uses water as the

working fluid. Typically, the gas turbine cycle is operated at a higher average temperature than the Rankine cycle.

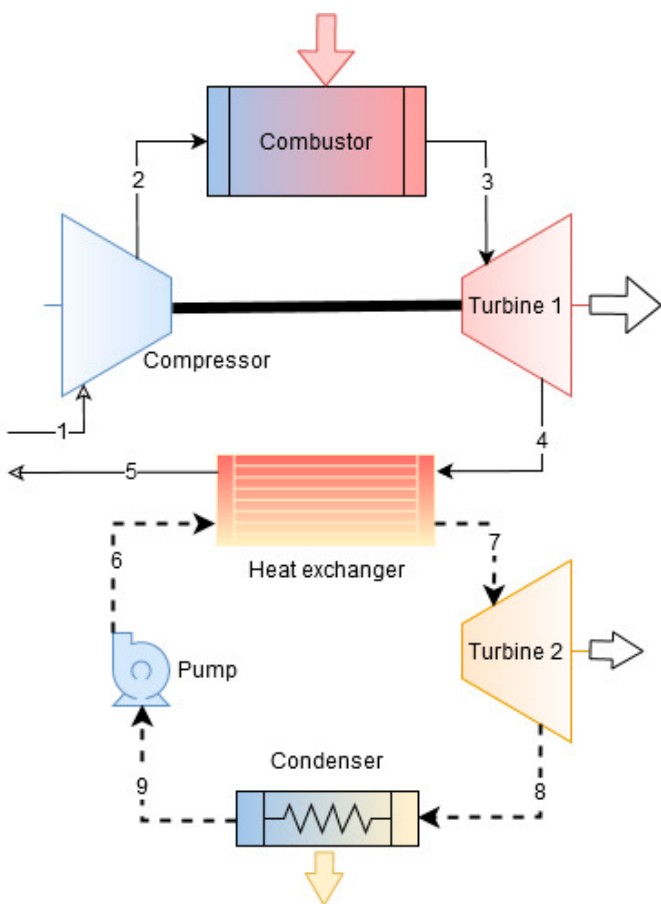

**Figure 1.** Schematic for combined cycle.

An ideal gas turbine cycle consists of a compressor, a combustion chamber, and a turbine with the following processes: isentropic compression process in the compressor (1–2), constant pressure heat addition process in the combustor (2–3), and isentropic expansion process in the turbine 1 (3–4). An ideal Rankine cycle consists of four process: isentropic compression process in the pump (9–6), constant pressure heat addition process in the evaporator (6–7), isentropic expansion process in the turbine (7–8), and constant pressure heat release process in the condenser (8–9). Note that in a combined cycle, an interconnecting heat exchanger is employed as the evaporator so that the heat discharged by the gas turbine cycle could be used partly or wholly as the input for the Rankine cycle to change water to vapor. Then, the vapor enters the turbine and expands isentropically in the turbine and generates work by rotating the turbine shaft until the pressure drops to the condenser pressure. Subsequently, exhaust vapor flows into the condenser where it is condensed to saturated liquid water. Then, liquid water enters the pump and is pressurized isentropically to the evaporating pressure and delivered to the interconnecting heat exchanger to utilize the energy discharged by the heat transfer from the gas turbine cycle. The following assumptions are made in the analysis:

a. Each component in the combined cycle is considered as a control volume at the steady state;

b. The compressor, turbines, pump, and heat exchangers are operated adiabatically;

c. Neglect kinetic and potential energy;

d. There is no pressure drop in the combustor, interconnecting heat exchanger, and condenser;

e.     An air-standard analysis is applied to the gas turbine cycle;
f.     It is assumed that the working fluid, air, has a constant specific heat ($c_P$) value of 300 K.

## 3. Method and Implementation

This section presents a process to implement the proposed method through a class assignment. Figure 2 illustrates the proposed method that would enable the students to develop a computational tool on their own into the course, with a comprehensive understanding of fundamentals, and evaluate its effectiveness for this study.

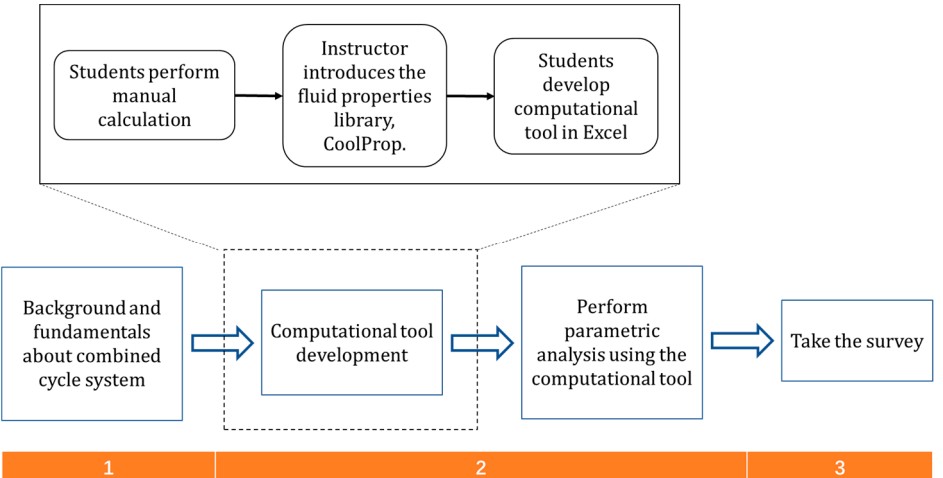

**Figure 2.** Method to implement the self-developed tool in the Power Generation Systems course.

Three assignments were given to the students. Through the first assignment, the students were asked to finish the calculation process by hand to demonstrate their understanding of the fundamentals. Then, in the second assignment, they were asked to develop the computational tool in Microsoft Excel and evaluate the system performance under varying operating conditions. In the last assignment, they were required to implement the computational tool to determine the effect of the evaporating pressure on the cycle power output and thermal efficiency, exergy efficiency, and exergy destruction in the cycle.

### 3.1. Assignment 1–Manual Calculation

The problem statement given to the students, and proper calculation steps to solve the problem are provided below. The students were required to use the thermal property tables in the textbook to evaluate gas and liquid properties to solve the problem.

*Problem description:* A combined gas turbine and Rankine cycle (as shown in Figure 1) has a heat input to the cycle of 74 MW and source temperature of 1500 K. In the compressor, air is compressed from 100 kPa, to 300 K, to 1200 kPa. After the heat addition process in the combustor, i.e., at the inlet of turbine 1, the air pressure and temperature are 1200 kPa and 1400 K, respectively. Then, air expands in the turbine to a pressure of 100 kPa. Subsequently, air flows through the interconnecting heat exchanger and leaves at a temperature of 400 K. At the Rankine cycle side, vapor enters turbine 2 at 8 MPa, 673 K, and then expands to the condenser pressure of 8 kPa. In the condenser, the vapor is condensed to saturated liquid water at 8 kPa. After that, water enters the pump and is pressurized from 8 kPa to 8 MPa. The isentropic efficiencies of the compressor, turbine 1, turbine 2, and the pump are 84%, 88%, 90%, and 80%, respectively. The ambient condition is set to 300 K and 100 kPa. Then, determine the net power output, thermal efficiency, the exergy efficiency, and the exergy loss of the combined cycle.

*Manual calculation steps:* The manual calculation process can be found in many thermodynamics textbooks. The step-by-step manual calculation process instructed to the

students is presented in the Appendix A. The equations used in the calculation are from Moran et al. [20]. As seen in the Appendix A, the manual calculation requires time and accuracy to obtain solutions, and the evaluation of the saturated water table multiple times.

### 3.2. Assignment 2–Computational Tool Development

This assignment was implemented using a student-centered approach of learning, in which the instructor facilitated student learning and guided the students during the tool development. By asking the students to develop a tool, the students were engaged in more active learning in the classroom. The problem statement that was given to the students, and an example of one student's developed computational tool, are provided below. The students were required to integrate an add-on module of the obtained fluid property to their self-developed computational tool.

*Problem description:* Based on the calculation process learned in the class, develop your own calculation sheet in Excel to determine the design and operational characteristics of gas turbine-based combined cycles, as shown in Assignment 1. Your calculation sheet should provide the following outputs:

- Unknown temperatures for gas turbine cycle;
- Heat input to the combustor ($\dot{Q}_{in}$);
- Mass flow rate of vapor cycle (Rankine cycle);
- Net work of gas turbine ($\dot{W}_{gas}$);
- Net work of vapor cycle ($\dot{W}_{vap}$);
- Overall system net work ($\dot{W}_{net}$);
- Overall system thermal efficiency (1st law efficiency);
- Exergy destruction of each component;
- Total system exergy destruction;
- Overall system exergy efficiency (2nd law efficiency).

*Example of student-developed computational tool*: The calculation process done by hand with table lookup for fluid properties is time-consuming, and does not provide the students with a clear understanding and visible variation trend when changing a certain parameter. To address this issue, the students were guided to develop their computational tool with an add-on module, i.e., CoolProp (A software program http://www.coolprop.org/), to obtain the properties of water and steam at different states. CoolProp is a C++-based library that provides pure and pseudo-pure fluid properties for 122 components and mixture properties [21]. With the tool, the students can vary the input data, components characteristics, and initial conditions to investigate the performance of the combined cycle system. One of the benefits of developing their own tool is that students gain a much better understanding of the fundamentals behind the cycle, as well as all the units involved in the different processes.

The data input tab and output result tab of the computational tool developed by a student are shown in Figures 3 and 4, respectively. The students can enter the known input data provided in Assignment 1 through the data input page illustrated in Figure 3 to perform an analysis of the combined cycle. According to the students' implementation of equations provided in Appendix A in the computational tool, the analysis results are outputted and displayed, as shown in Figure 4. It can be observed from the figures that for students to use the tool, they only need to input the known data for the combined cycle, and the tool will quickly provide the desired outcomes for the problem, such as the net power output, thermal efficiency, and exergy efficiency of the combined cycle. As mentioned before, the computational tool uses CoolProp to obtain the properties of water. Note that there are small differences between the CoolProp values and Table values, which cause a little difference in the calculation results.

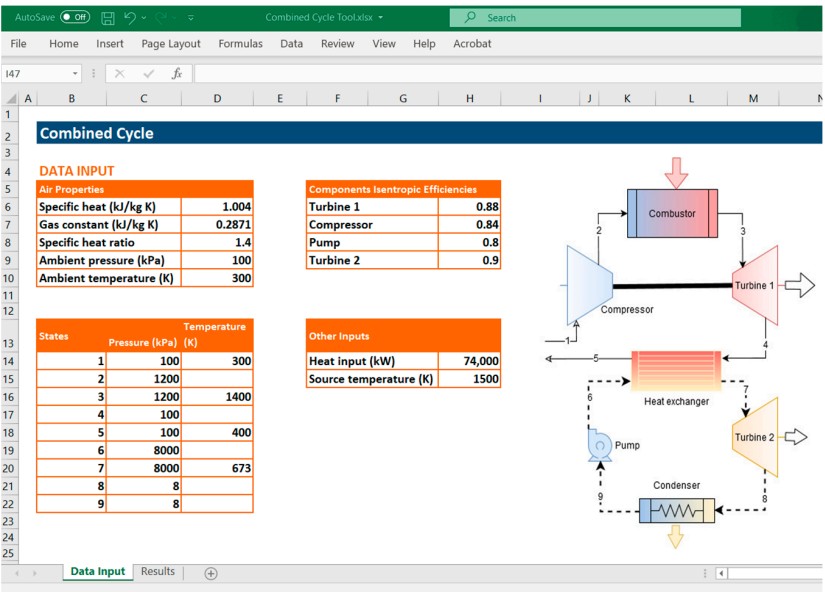

**Figure 3.** Screenshot of student-developed computational tool in Excel—data input.

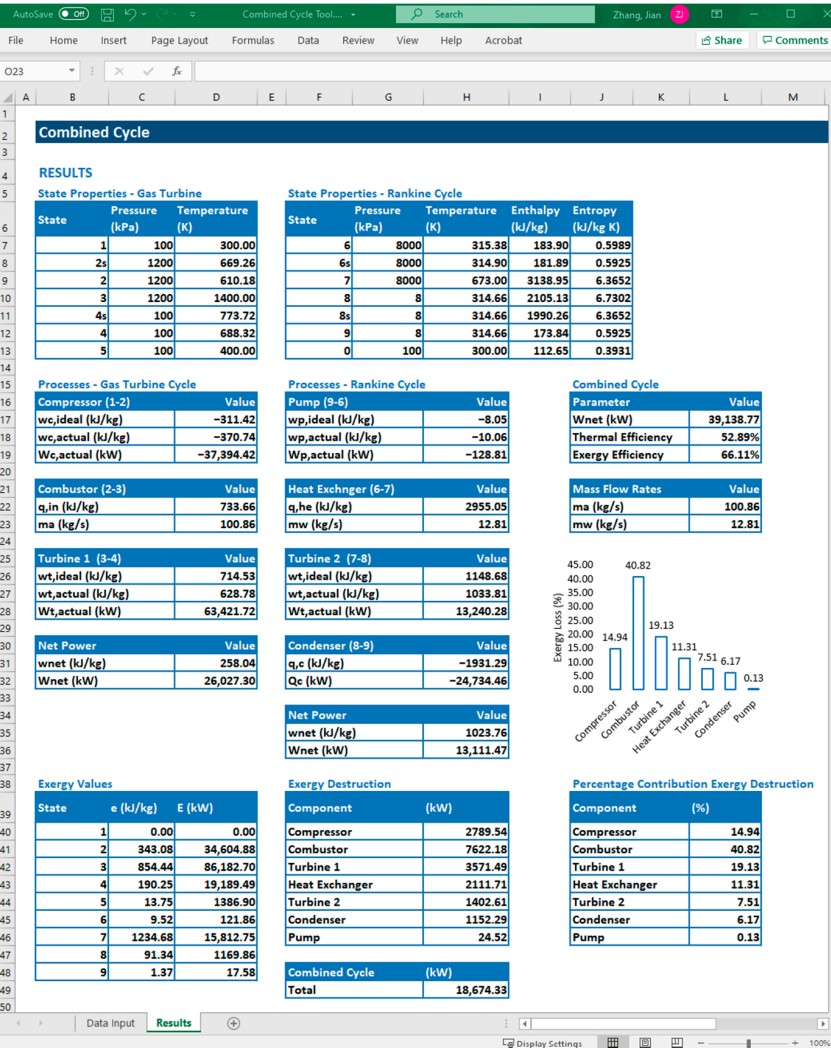

**Figure 4.** Screenshot of student-developed computational tool in Excel—output results.

Figure 4 shows that the net power output, thermal efficiency and exergy efficiency of the combined cycle are determined to be 39.14 MW, 52.89%, and 66.11%, respectively. In addition, the students can quickly perform an exergy analysis of the combined cycle using the computational tool. The exergy rate at each state and the exergy destruction in each component are also presented in Figure 4. Three assignments were given to the students. Through the first assignment, the students were asked to finish the calculation process by hand to demonstrate their understanding of the fundamentals. Then, in the second assignment, they were asked to develop the computational tool in Microsoft Excel and evaluate the system performance in varying operating conditions. In the last assignment, they were required to implement the computational tool to determine the effect of evaporating pressure on the cycle power output and thermal efficiency, exergy efficiency, and exergy destruction in the cycle. From this assignment and example tool shown in Figures 3 and 4, it can be clearly observed that the tool significantly simplifies the calculating procedures, including the property evaluations of the working fluid. In addition, it can be used to investigate the influence of changing some parameters over the performance of the combined cycle, as illustrated in Assignment 3 in the next section.

### 3.3. Assignment 3

*Problem description:* Using all the input values, except for evaporating pressure $P_7$, for a combined cycle from Assignment 1, vary $P_7$ from 2 MPa to 8 MPa to determine the effect on the cycle power output and thermal efficiency, exergy efficiency, and exergy destruction in the cycle.

*Enhanced parametric study process using a computational tool:* This is an example where the computational tool is very useful to perform the parametric analysis. With the computational tool developed by themselves, the students gain the ability to see immediate results to variations in input design conditions, as well as with different parameters. The manual calculations in the parametric study can be time-consuming and tedious using the formulations described in Section 3.1.

Figure 5 show the plots based on the results provided by the tool for different evaporating pressures. As can be seen in these figures, both the net power output (Figure 5a) and the thermal efficiency (Figure 5b) increase with the increase in the evaporating pressure ($P_7$). Moreover, the slope of both curves decreases as the evaporating pressure increases. Figure 6 presents the influence of varying the evaporating pressure on the exergy destruction of the combined cycle and on the heat exchanger, turbine 2, and condenser. It can be seen from Figure 6 that the variation in the total exergy destruction is mainly influenced by the exergy destruction in the heat exchanger, which decreases with the increase in the evaporating pressure.

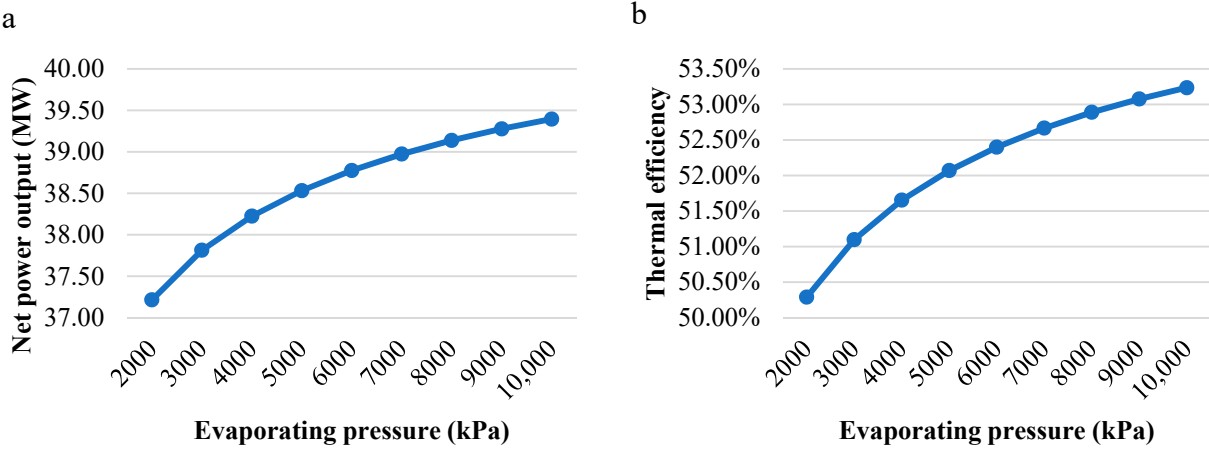

**Figure 5.** Influence of varying evaporating pressure ($P_7$) on the cycle net power output (**a**) and thermal efficiency (**b**).

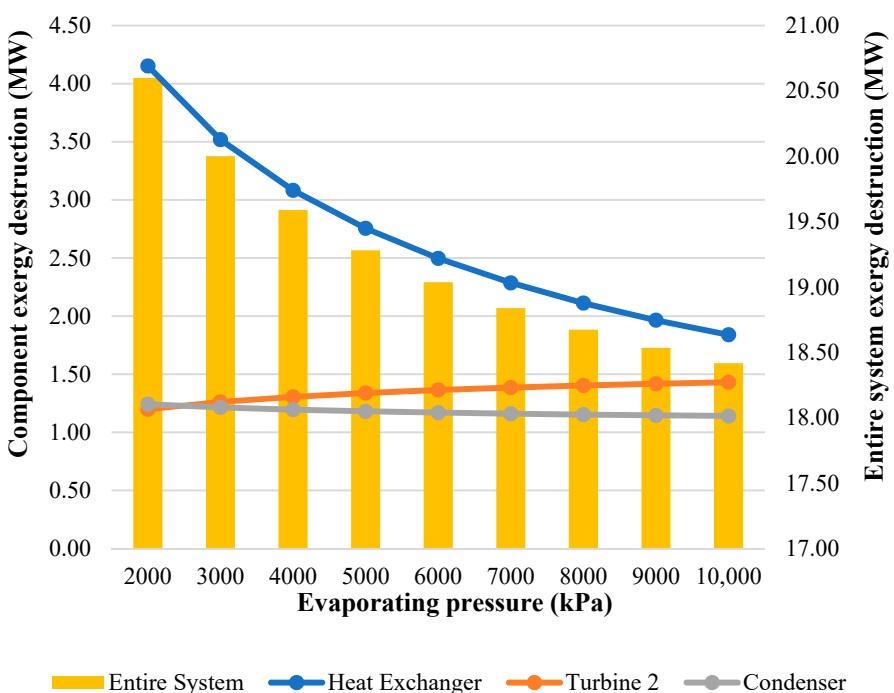

**Figure 6.** Influence of varying evaporating pressure ($P_7$) on the exergy destruction

## 4. Survey Results and Discussion

The students were asked to use the self-developed tool to solve several homework problems after they had the opportunity to solve them manually. After using the tool and evaluating the effectiveness of the tool, the students were given an optional survey asking their opinion during the Spring 2020 semester, and 13 students out of 15 (87% of the class) submitted responses. The questions are compiled in Table 1. Questions 1–4 are 1–5 Likert Scale responses, with 5 representing strongly agree, 4 agree, 3 neither agree or disagree, 2 disagree, and 1 strongly disagree, while Q5 is for students to provide comments. The average responses to Likert Scale Questions 1–4 are compiled and displayed in Figure 7.

**Table 1.** Survey questions administered to the class.

| Question | Statement |
|----------|-----------|
| Q1 | The implementation of a calculation tool in a computer software tool enhances my understanding of the physics behind the combined cycle. |
| Q2 | The implementation of a calculation tool in a computer software tool provides a convenient method to investigate or visualize the variation of the output due to changes in the input. |
| Q3 | The implementation of a calculation tool enables me to focus more on the design of the cycle, rather than the calculation. |
| Q4 | I am interested in developing and using more tools like this one to solve other cycles or problems. |
| Q5 | In addition to those listed above, what are the benefits by developing and using the calculation tool compared to conducting calculations by hand? |

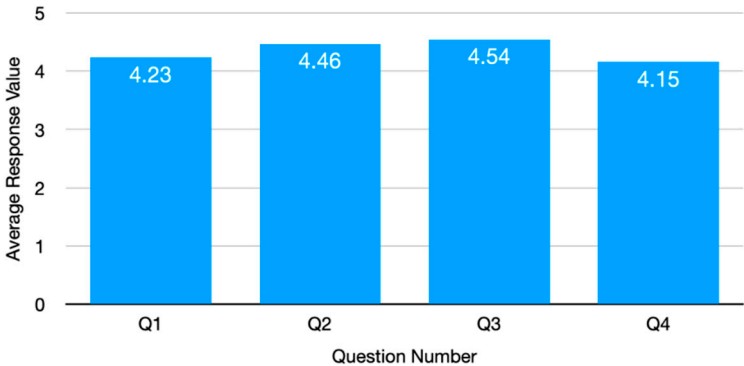

**Figure 7.** Average Likert response for Questions 1 through 4.

The results presented in Figures 7 and 8 indicate that the students viewed the tool implementation positively. The average for Q1 was 4.23, and 11 out of 13 students (84.6% of the students) responded "strongly agree" or "agree" to this question. Therefore, it implies that the implementation of a calculation tool in a computer software tool helps the students focus on the physics behind the combined cycle, which help them enhance their understanding. The average of Q2 was 4.46, and 12 out of 13 students (92.3% of the students) responded "strongly agree" or "agree" to this question. This is an indication that the implementation of a calculation tool in a computer software tool provides a convenient method for the students to investigate or visualize changes in some parameters that would affect the system performance. Similarly, the average of Q3 was 4.53, the highest, and 12 out of 13 students (92.3% of the students) also responded "strongly agree" or "agree" to this question. This validates one of the main objectives of the tool, which is allowing the students to focus more on the design of the cycle rather than the calculations. The average of Q4 was 4.15, the lowest. Of 13 students, 10 (76.9% of the students) responded "strongly agree" or "agree," 2 out of 13 students (15.4%) responded "neutral," and 1 out of 13 students (7.7%) responded "disagree" to this question. Of the students, 76.9% are interested in developing and using more tools like this one to solve other cycles or problems. However, it is evident that some of the students, even though they like to use similar tools, are not interested in developing their own tools.

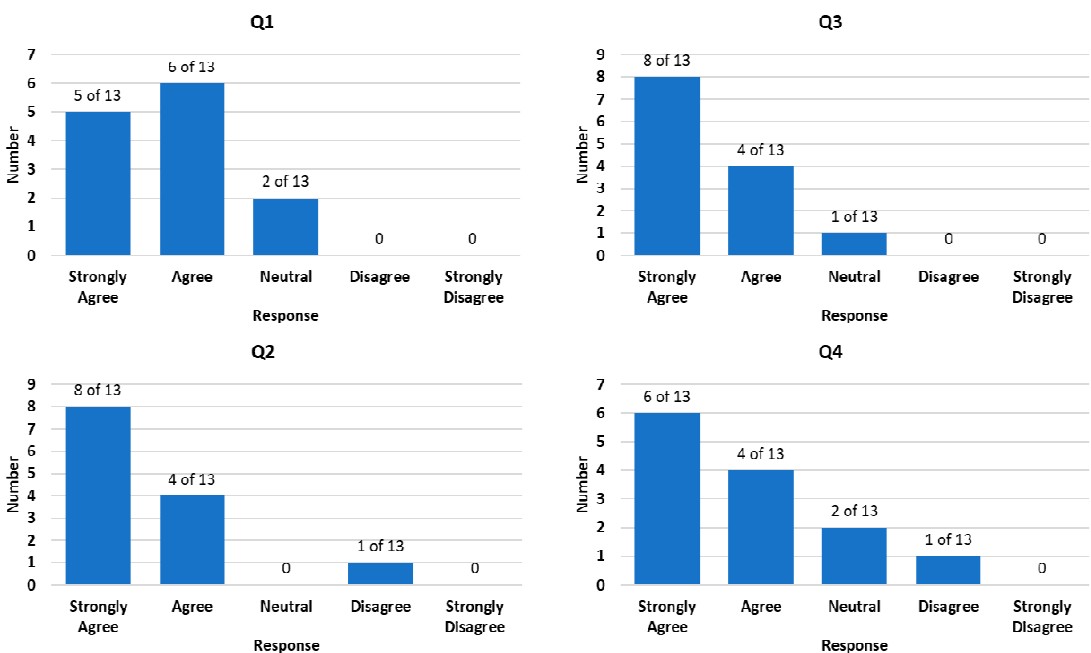

**Figure 8.** Percent of students per question and response.

Table 2 includes some of the comments (unedited) received from the students for Question 5: "In addition to those listed above, what are the benefits by developing and using the calculation tool compared to conduct calculations by hand?"

**Table 2.** Responses for Question 5.

| Student Comments |
| --- |
| • Not having to look up values in tables is very nice. Quickly changing the values to see how it affects the cycle is also nice. |
| • There can be less hand derived errors in the calculations in addition to a larger array of calculations that can be calculated at the same time. |
| • It eliminates the need to perform the same calculations for the same cycle. |
| • Saves time of repetitive calculations. |
| • Less time being caught up in the math and more time focusing on the actual material we learn in class. |
| • It is more efficient and less time consuming. |
| • It helps understand the calculations and having the tool helps see the different outcomes of different variables and input. |
| • It is a lot quicker and easy to use. |
| • Aside from the sensitivity analysis already mentioned, it also exposes us to the computer tools that we might use in industry. |

The responses from Question 5, and additional feedback from students and class instructors, demonstrate that the use of these tools enhances the student experience in the power generation systems class. Using the proposed tool, the students gain the ability to evaluate the problem in a different way, allowing them to see immediate results in response to variations in input design conditions, as well as different parameters. Therefore, from the feedback received from students, the proposed pedagogical methodology keeps them more engaged in active learning in the classroom.

From the instructor's perspective, a comparison between the evaluation of this topic with and without the tool implementation shows the advantages of using the proposed tool. Without the tool, the homework problems were typically limited to one set of conditions to determine the combined cycle first and second law efficiency, as well as exergy analysis. However, with the tool, a more comprehensive problem can be assigned in which students are asked to solve the same problem with the addition of changing different variables to study how they will affect the combined cycle performance. This can give the students a much better understanding of the problem and changes in the combined cycle system performance based on the parametric evaluations. The proposed student-centered learning approach facilitates student learning and gives the instructor the opportunity to assign more comprehensive and challenging problems to students and encourage more discussion and class participation. This is well in agreement with other studies, e.g. [5–8,12,13,18,19], wherein similar approaches utilizing computational tools were incorporated in their classrooms.

## 5. Conclusions

This paper presented a methodology to improve the student learning of energy systems by requiring students to implement a self-developed computational tool and utilize the tool to enhance their learning experience in their course assignments. This study was designed to show that the self-developed computational tool can help students reduce the calculation steps and simplify the process to focus more on analyzing the results and understanding the fundamental concepts of energy systems. A case study of implementing the proposed methodology in an energy-focused engineering course was provided to illustrate the advantages of the self-development of computational tools in their course learning. With the computational tool, the students could quickly determine the results due to changes in input data and/or system parameters. Feedback from students indicated that the self-development of the computational tool enhanced the understanding of the physics behind the course materials and improved the enjoyment of the students' learning

experience. The students were interested in developing and using more tools like the one described in this paper to solve other problems in different subjects. Feedback from the instructor indicated that the self-developed tool facilitated effective student learning and gave the instructor the opportunity to assign more comprehensive and challenging problems while maintaining the students' engagement. Therefore, it can be concluded that the self-development and use of the tool can significantly improve the students' learning experience in energy-related courses, make courses more dynamic, and motivate the students to learn the material more iteratively.

It should be noted that a limited number of students (i.e., 15 students) from a single course provided feedback on the presented method. Based on this study, future work can be extended to include more samples from various engineering courses.

**Author Contributions:** Conceptualization, J.Z., H.C. and P.J.M.; methodology, J.Z., H.C. and P.J.M.; software, J.Z.; validation, J.Z.; formal analysis, J.Z.; investigation, J.Z.; data curation, J.Z.; writing—original draft preparation, J.Z.; writing—review and editing, H.C. and P.J.M.; visualization, J.Z.; supervision, H.C. and P.J.M. All authors have read and agreed to the published version of the manuscript.

**Funding:** This research received no external funding.

**Institutional Review Board Statement:** Not applicable.

**Informed Consent Statement:** Not applicable.

**Data Availability Statement:** The data presented in this study are available within this article.

**Conflicts of Interest:** The authors declare no conflict of interest.

## Appendix A

To solve this example manually, the properties at each state as shown in Figure 1 must be determined.

(a) The air temperature at the exit of the compressor for an isentropic process can be determined by:

$$\frac{T_{2s}}{T_1} = \left(\frac{P_2}{P_1}\right)^{\frac{k-1}{k}} \tag{A1}$$

where $k$ is the specific heat ratio which equals the ratio of the specific heat at a constant pressure to the specific heat at a constant volume. Based on the input data, $\frac{P_2}{P_1} = 120$, then $T_{2s}$ is 610.18 K. Then, the $T_2$ can be determined using the following equation, which is 669.26 K.

$$\eta_c = \frac{w_{c,s}}{w_{c,a}} = \frac{h_1 - h_{2s}}{h_1 - h_2} = \frac{c_p(T_1 - T_{2s})}{c_p(T_1 - T_{2s})} \tag{A2}$$

where $w_{c,s}$ and $w_{c,a}$ are the specific work of the isentropic and actual processes, respectively. $h$ and $T$ are the specific enthalpy and temperature at each state, respectively. In this model, the working fluid, air, is modeled as the ideal gas with constant specific heat $c_p$. Note that Process 1 to 2 indicates an isentropic process, thus the entropy satisfies the following relationship:

$$s_1 = s_{2s} \tag{A3}$$

(b) The air temperature at the exit of the turbine 1 for an isentropic process can be determined in the same way:

$$\frac{T_{4s}}{T_3} = \left(\frac{P_4}{P_3}\right)^{\frac{k-1}{k}} \tag{A4}$$

$$\eta_t = \frac{w_{t,a}}{w_{t,s}} = \frac{h_3 - h_4}{h_3 - h_{4s}} = \frac{c_p(T_3 - T_4)}{c_p(T_3 - T_{4s})} \tag{A5}$$

where $w_{t,s}$ and $w_{t,a}$ are the specific work of turbine 1 for the isentropic and actual processes, respectively. Similar to the compression process, Process 3 to 4 s is an isentropic process:

$$s_3 = s_{4s} \tag{A6}$$

According to the calculation results, $T_{4s}$ and $T_4$ are 668.32 K and 773.72 K, respectively. By now, the temperature of air at each state is determined, which can be used to evaluate the performance of the gas turbine cycle since the air is considered as the ideal gas in this model.

(c)   The mass flow rate of the air is then calculated to be 100.86 kg/s by:

$$\dot{m}_a = \frac{74\ MW}{q_{in}} = \frac{74\ MW}{C_p(T_3 - T_2)} \tag{A7}$$

(d)   Along with the equation below, the net power output of the gas turbine cycle is 26.03 MW.

$$w_{net,GT} = w_{c,a} + w_{t,a} \tag{A8}$$

(e)   At the Rankine cycle side, the specific enthalpy and specific entropy are required for each state. The properties are determined using the vapor tables. States 7 and 9 can be fully determined using the given information. At State 9, the water is saturated liquid (quality of 0), and the pressure is 8 kPa. Therefore, the properties can be obtained from the saturated water tables. To determine States 6 and 8, two ideal States 6s and 8s were introduced, which converts Process 9 to 6s and 7 to 8s as two isentropic processes. Finally, all the states for the Rankine cycle can be completely determined, though only specific enthalpy and entropy are needed. The properties obtained from the water tables for each state are summarized in Table A1.

**Table A1.** Specific enthalpy and entropy at each state in the Rankine cycle (Table values).

| State Points No. | Enthalpy (kJ/kg) | Entropy (kJ/kg K) |
|:---:|:---:|:---:|
| 6 | 184.46 | 0.5990 |
| 6s | 182.34 | 0.5926 |
| 7 | 3138.30 | 6.3634 |
| 8 | 2105.33 | 6.7299 |
| 8s | 1990.55 | 6.3634 |
| 9 | 173.88 | 0.5926 |

(f)   Subsequently, these properties can be used to evaluate the performance of the Rankine cycle. Using the equation below, the mass flow rate of water can be found to be $\dot{m}_w = 12.81$ kg/s.

$$\frac{\dot{m}_a}{\dot{m}_w} = \frac{h_7 - h_6}{c_p(T_4 - T_5)} \tag{A9}$$

(g)   Along with the following equations, the net power output of the Rankine cycle can be calculated to be 13.10 MW.

$$\eta_t = \frac{w_{t,a}}{w_{t,s}} = \frac{h_7 - h_8}{h_7 - h_{8s}} \tag{A10}$$

$$\eta_p = \frac{w_{p,s}}{w_{p,a}} = \frac{h_9 - h_{6s}}{h_9 - h_6} \tag{A11}$$

$$w_{net,R} = w_{p,a} + w_{t,a} \tag{A12}$$

(h) Using the equations below, the thermal efficiency of the combined cycle can be determined as 52.87%.

$$w_{net} = w_{net,GT} + w_{net,R} \tag{A13}$$

$$\eta_t = \frac{w_{net}}{q_{in}} \tag{A14}$$

(i) In addition, the exergy rate at each state can be determined based on Equations (A15)–(A17). The results are summarized in Table A2.

**Table A2.** Exergy rate at each state.

| State Points | Exergy Rate (MW) |
|:---:|:---:|
| 1 | 0.00 |
| 2 | 34.60 |
| 3 | 86.18 |
| 4 | 19.19 |
| 5 | 1.39 |
| 6 | 0.13 |
| 7 | 15.82 |
| 8 | 1.18 |
| 9 | 0.02 |

The exergy flow at each point (or state), shown in Figure 1, can be expressed as:

$$E_i = (h_i - h_0) - T_0(s_i - s_0) \tag{A15}$$

where $E$ indicates the exergy and $s$ is the entropy; $i$ implies each point in Figure 1, that is, State 1 to 9; $T_0$, $h_0$, and $s_0$ are the ambient temperature, enthalpy and entropy of working fluid at ambient temperature, respectively.

For air, which is modeled as the ideal gas in the analysis, the exergy flow is calculated by the following equation:

$$E_i = c_p(T_i - T_0) - T_0 \left( c_p ln \frac{T_i}{T_0} - R ln \frac{P_i}{P_0} \right) \tag{A16}$$

where $T$ and $P$ are the temperature and pressure, respectively, and $R$ is the gas constant.

Once the exergy flow is determined, the exergy rate at each state can be expressed as:

$$\dot{E}_i = \dot{m} E_i \tag{A17}$$

where $\dot{m}$ represents the mass flow rate of the working fluid.

(j) Based on the exergy rate calculated in last step, exergy destruction in each component as well as for the entire system can be estimated using Equations (A18)–(A27). The results are listed in Table A3.

**Table A3.** Exergy destruction in each component and entire system.

| Component | Exergy Destruction (MW) |
|:---:|:---:|
| Compressor | 2.79 |
| Combustor | 7.62 |
| Turbine 1 | 3.57 |
| Heat Exchanger | 2.11 |
| Turbine 2 | 1.41 |
| Condenser | 1.16 |
| Pump | 0.02 |
| Entire System | 18.69 |

For the combined cycle, heat transfer and work are involved in many processes. The exergy rate due to heat transfer and work are described by Equations (A18) and (A19), respectively.

$$\dot{E}_Q = \dot{Q}\left(1 - \frac{T_0}{T_H}\right) \tag{A18}$$

$$\dot{E}_W = \dot{W} \tag{A19}$$

With the exergy rate determined, the exergy destruction or exergy loss in each component, i.e., for each process in the cycle, can be calculated as below.

*Compressor (Process 1–2):*

$$\Pi_I = \dot{E}_1 + \dot{E}_{WC} - \dot{E}_2 \tag{A20}$$

where $\dot{E}_{WC}$ is the exergy rate due to the compressor work.

*Combustor (Process 2–3):*

$$\Pi_{II} = \dot{E}_2 + \dot{E}_{in} - \dot{E}_3 \tag{A21}$$

where $\dot{E}_{in}$ is the exergy rate due to the heat transfer into the combustor.

*Turbine 1 (Process 3–4):*

$$\Pi_{III} = \dot{E}_3 - \dot{E}_{WT1} - \dot{E}_4 \tag{A22}$$

where $\dot{E}_{WT1}$ is the exergy rate due to work produced by turbine 1.

*Heat exchanger (Process 4–5, 6–7):*

$$\Pi_{IV} = \dot{E}_4 + \dot{E}_6 - \dot{E}_7 - \dot{E}_5 \tag{A23}$$

*Turbine 2 (Process 7–8):*

$$\Pi_V = \dot{E}_7 - \dot{E}_{WT2} - \dot{E}_8 \tag{A24}$$

where $\dot{E}_{WT2}$ is the exergy rate due to work produced by turbine 2.

*Condenser (Process 8–9):*

$$\Pi_{VI} = \dot{E}_8 - \dot{E}_{out} - \dot{E}_9 \tag{A25}$$

where $\dot{E}_{out}$ is the exergy rate due to the heat transfer out of the condenser.

*Pump (Process 9–6):*

$$\Pi_{VII} = \dot{E}_9 + \dot{E}_{WP} - \dot{E}_6 \tag{A26}$$

where $\dot{E}_{WC}$ is the exergy rate due to the pump work.

Finally, the exergy destruction for the entire system is the summation of exergy loss in each component:

$$\Pi_{sys} = \sum_{j=I}^{VII} \Pi_j \tag{A27}$$

(k) The cycle exergy efficiency can be determined using Equation (A28) as 66.09%.

$$\eta_{II} = \frac{W_{net}}{E_{in}} \tag{A28}$$

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
