# Peer review of "Improving Student Learning of Energy Systems through Computational Tool Development Process in Engineering Courses"

_sustainability, doi:10.3390/su13020884_

Round 1

Reviewer 1 Report

Lines 93 to 99 are from the template?

It is difficult to find out which is the real aim of the paper. It is mixed with the state of the art. I think that the aim of the paper should be found easily in the introduction.

Nor the method used to analyse the data can't be found easily. No discussion of the results.

The conclusions refer a method (case study) that it is not presented throughout the paper. They do not refer to the objective of the paper (I don’t really know which is the aim after the reading)

Figure 1 and its explanations should be in another section where the experience and phases of the study are explained.

SECTION 2: It starts with the instructions of the template?

I would better explain the relevance of the figures 3 and 4.

Section 4: 

Results are regarding only 15 students and have been analyzed using quantitative analysis when the sample was very small. They should have used with such a small sample, at least, a mixed method for analysis, a mixed method with qualitative analysis could have been more rich in their results. Any how the methodology is not well exposed, nor the discussion of the results.

In that case I consider that quantitative methodology alone doesn’t give relevant results (comments should be qualitatively analyzed to understand the results better). I would never give the results as percentages but something similar to:5 of 15 strongly agree, 7 of 15 agree.

Reviewer 2 Report

An interesting case study that illustrates students' adherence to active and design methodologies.
It would be expected that a computational tool would be well received by students of an engineering course. It would be interesting to replicate this study with students from courses in other scientific areas.

This manuscript, like others, confirms that student-centered and project-based teaching methods are effective in the teaching and learning process.

The manuscript is too based on the computational tool, when the most important is the applied pedagogical methodology. The last sentence of the introduction shows this well.

The problem is described, as well as how to solve it.

The study was assigned to a small group of students (15).

It would be important to apply this pedagogical method to students from other courses and to do a comparative analysis.

Despite the limitations pointed out, the study reports an exercise in applying an active learning process and shows its positive effects. Emphasis should also be placed on the pedagogical method and not on the computational tool.

The objectives and methodology are correctly defined.

Round 2

Reviewer 1 Report

The content is not well contextualized with respect to previous and present theoretical background.

You say that the aim is “aims to implement and evaluate a pedagogical method” but the conclusions are referred to a different aim (“This study was designed to demonstrate that”). If you want to evaluate a pedagogical method, you should compare it with the literature in the discussion section.

The conclusion should help to understand why your paper is important. However, your conclusion is merely a summary of your key points.

You say that “This study was designed to demonstrate that”. I don’t think you can demonstrate something with just 15 students. May be another verb should be used.

I think that your paper is more focused in the method used to implement the tool than in the students itself. May be a discussion focused on that should be better supported by some references, so you can show what new knowledge you have contributed. May be this can be added in the discussion section. I don’t think it is a case study. Conclusions my be should be referred to the developed tool on “energy education” more than to the students.

Also, I cannot find any limitations of your study in the conclusions and I think it is important to point out the implications of a study or make any recommendations for future work in the topic.

Author Response

 Responses to Editor and Reviewer’s Comments

Thank you for the reviews on our paper. We have revised the manuscript per the reviewer’s comments and have outlined how we addressed each comment below. We found the reviews of our manuscript insightful and helpful in improving the paper. We extend our thanks to the reviewer and the editorial team.  

Reviewer 1

Comment 1: The content is not well contextualized with respect to previous and present theoretical background.

Response: Thank you for your comment. We are not sure exactly what the reviewer is referring for “previous and present theoretical background.” It would be great if you could be more specific what content is not well contextualized.

Comment 2: You say that the aim is “aims to implement and evaluate a pedagogical method” but the conclusions are referred to a different aim (“This study was designed to demonstrate that”). If you want to evaluate a pedagogical method, you should compare it with the literature in the discussion section.

Response: The term “evaluate” was removed in compliance with the reviewer’s comment.  The revised sentence in the Introduction reads:

This paper aims to present a pedagogical method, using a student-centered learning approach, to improve the student learning of energy systems through a class assignment implementing a self-developed computational tool using Microsoft Excel in an energy-related engineering course, called Power Generation Systems.

            Similarly, the following sentence was stated in the Conclusions section to clarify this:

This paper presented a methodology to improve the student learning of energy systems by requiring students to implement a self-developed computational tool and utilizing the tool to enhance their learning experience in their course assignments.

Comment 3: The conclusion should help to understand why your paper is important. However, your conclusion is merely a summary of your key points.

Response: The following statement has been added in the Conclusions section to emphasize the important of this study as suggested by the reviewer:

Therefore, it can be concluded that the self-development and use of the tool can significantly improve the students’ learning experience in energy-related courses, make courses more dynamic, and motivate the students to learn the material more iteratively.

Comment 4: You say that “This study was designed to demonstrate that”. I don’t think you can demonstrate something with just 15 students. May be another verb should be used.

Response: The manuscript has been edited as suggested by the reviewer, i.e., and replace the term “demonstrate” was replaced with “show.” 

Comment 5: I think that your paper is more focused in the method used to implement the tool than in the students itself. May be a discussion focused on that should be better supported by some references, so you can show what new knowledge you have contributed. May be this can be added in the discussion section. I don’t think it is a case study. Conclusions my be should be referred to the developed tool on “energy education” more than to the students.

Response: Please note that the tool was developed by the students. Furthermore, they used the self-developed tool in their assignments and provided feedback on their effectiveness on their learning outcomes. Therefore, we believe that the conclusions should be referred to their feedback and learning from the method implemented in the classroom.   

Comment 6: Also, I cannot find any limitations of your study in the conclusions and I think it is important to point out the implications of a study or make any recommendations for future work in the topic.

Response: Thank you for your comment. The following statements are added at the end of Conclusions:

It should be noted that limited number of students (i.e., 15 students) from a single course provided feedback on the presented method. Based on this study, future work can be extended to include more samples from various engineering courses.  

Round 3

Reviewer 1 Report

It has been improved. However, I still think that some more references are needed to support the discussion of the results and conclusions

Author Response

Thank you for the reviews on our paper. We have revised the manuscript per the reviewer’s comments and have outlined how we addressed each comment below. We found the reviews of our manuscript insightful and helpful in improving the paper. We extend our thanks to the reviewer and the editorial team.  

Reviewer 1

Comment: It has been improved. However, I still think that some more references are needed to support the discussion of the results and conclusions.

Response: Thank you for your comment. The following papers have been added to the manuscript as suggested by the reviewer:

  1. M. Maixner, “Excel™ Analysis of Combined Cycle Power Plant,” in Proc. of the 2005 American Society for Engineering Education Annual Conference & Exposition, Portland, OR, June 2005.
  2. A. A. Knizley and P. J. Mago, "Implementation of Computational Tools in Energy-Related Mechanical Engineering Courses," in 2014 ASEE Southeast Section Conference, Macon, GA, March 2014.
  3. A. A. Knizley and P. J. Mago, "Computational Tools to Enhance the Study of Gas Power Cycles in Mechanical Engineering Courses," in 2015 ASEE Southeast Section Conference, Gainesville, FL, April 2015.

In addition, the following statements were added to support the discussion of the results and conclusions.

Knizley and Mago [18,19] demonstrated that utilizing Excel spreadsheet-based tools in the classroom can promote a student-centered learning environment and improve the student understanding of class materials and teaching efficiency in engineer courses. 

This is well in agreement with other studies, e.g., [5-8,12,13,18,19], where similar approaches utilizing computational tools were incorporated in their classrooms.